# How Does Paradoxical Leadership Affect Employees’ Voice Behaviors in Workplace? A Leader-Member Exchange Perspective

**DOI:** 10.3390/ijerph17041162

**Published:** 2020-02-12

**Authors:** Ying Xue, Xiyuan Li, Hao Liang, Yuan Li

**Affiliations:** Economics and Management School, Wuhan University, Wuhan 430072, Hubei, China

**Keywords:** paradoxical leadership, voice behavior, psychological safety, self-efficacy, leader-member exchange theory, organizational citizenship behavior

## Abstract

We theorized and tested a leader-member perspective beyond the existing studies in paradoxical leadership and employee voice behavior. We proposed that paradoxical leadership influences employees’ voice behavior through psychological safety and self-efficacy. We also theorized that team size influences an extent to which the subordinates internalize their self-efficacy and psychological safety to exhibit proactive behavior. In a longitudinal study conducted on 155 subordinates and 96 supervisors in China, we found that when leaders adopt paradoxical behavior, employees are more likely to engage into promotive voice behavior; however, employees’ prohibitive voice behavior is reduced when their leaders adopt paradoxes in leadership behavior. Additionally, psychological safety mediates the relationship between paradoxical leadership and promotive voice behavior. Further, team size has significant interaction effects with psychological safety on promotive voice behavior.

## 1. Introduction

The contemporary business environment has become more dynamic than ever, with plenty of challenges and opportunities for leaders. With such dynamism, leaders need to manage paradoxes that typically denote a challenging tension, yet with interrelated elements [1]. Pearce et al. [2] performed a qualitative thematic analysis that was structured around meta-paradox or higher-level paradox that leaders encounter while adopting different forms of leadership such as shared, situational, or paradoxical. It was found that paradoxical leaders were the most effective when dealing with organizational paradoxes that were associated with balancing short-term and long-term goals. The logical implications of the findings of Pearce et al. [2] suggest that by utilizing the capacity to view higher-level paradox, i.e., challenging situations, leaders tend to boost the organizational and well as individual performance. Further, the individual and competing leadership paradoxes may encourage or restrict voice expression and speaking up behavior of employees [3]. Not surprisingly, the literature offers evidences on the link between different leadership styles and voice behavior (see Table 1). However, there is no explicit evidence on how paradoxical leadership reflects on efforts to productively utilize employees’ experiences and knowledge by influencing their voice behavior.

Leadership has been considered to be one of the key factors influencing voice behavior related risks and benefits [4]. Employee voice concerns their proactive, willing, informal and upward communication in relation to their ideas and solutions to the work-related issues. The employees feel risky when involved in groups or when they work under pressure because speaking up in such circumstances may question their status quo [5]. The voice behavior also involves costs such as humiliation, negative impact to social standing, demotion, and termination, all of which restrain the employees from speaking up [6,7,8]. At the same time, many evidences suggested positive performance outcomes related to voice behavior [9,10,11]. Thus, it is challenging for leaders as how to tackle voice behavior positively and at the same time, how to sustain creativity and innovation at the workplace.

The paradoxical leadership behavior is structured around the Yin and Yang philosophy, stating that all universal things exist as a balance of contradictory and inseparable opposites. Although not all studies on paradoxical leadership support this philosophy, yet in terms of paradoxical thinking, the interactive effects of visionary and empowering leadership engender more positive influence on followers’ behavioral and work outcomes [12]. In the context of organizational teams, numerous empirical studies have inferred that innovative performance is not easily achieved by diversified teams [13,14,15] and the underlying reason is that such teams face the paradox of differentiation–integration [16]. The paradoxical leadership traits behaviorally integrate and accept opposite demands in parallel to gain from the intent behind the paradox [17,18]. Nevertheless, leaders’ assumption of employees’ willingness and ability to engage in voice behavior is potentially dysfunctional for their learning as it may belie paradoxes and tensions [19]. This calls for more understanding on the link between paradoxical leadership and voice behavior.

The evidences from existing literature have suggested that higher the self-efficacy, the higher the confidence in the ability to make decisions and complete a task successfully will be [20,21] (see Table 1). Similarly, the rationale behind psychological safety is that organizational members believe that failures are a part of learning and speaking up, as well as inquiring about anything is not subject to any form of repercussion from their leaders or other members [22]. Authors have suggested that by targeting role modeling behaviors, leaders can increase the self-efficacy of their followers [23,24], and by creating a supportive environment for employees to balance tensions in creativity, the paradoxical leadership behavior eventually strengthens self-efficacy of employees and psychological safety at the workplace [22]. However, the degree to which self-efficacy is formed in an organization is influenced both by paradoxical leadership behavior and the freedom of voice. Nonetheless, a leader will be better able to recognize the team members in small teams through proper leader member exchange communication [25]. As such, it will be expected from the team members to engage in more self-driven behavior and thus, have higher self-efficacy levels [5]. 

This paper aims to analyze the impact of paradoxical leadership behavior on voice behavior of employees through self-efficacy and psychological safety. This research also theorized that leader-member exchange behavior between paradoxical leaders and employees influences voice behavior to an extent to which employees internalize their psychological safety and self-efficacy to promote their voice behavior. The existing studies have suggested that paradoxical leadership behavior has a positive moderating impact on voice behavior. Although earlier studies have been performed around this relationship, no direct relationship has been established. The current study proposes to fill this gap by finding a direct relationship.

It is interesting to note that while leader-member exchange theory has produced a successful and growing area of research in the empirical area, more theoretical interpretations are needed [18,32]. Although it is widely studied that leader-member exchange leads to higher work outcomes and subordinate performance, the understanding on “how” and “why” of these relationships are less explored in the literature [18]. Since the essence of paradoxes involves opposites and dealing with them separately, a theoretical lens of leader-member exchange theory can generate better insights on the interpretation of results of this study. Considering the fact that leader-member exchange theory has been widely generalized by scholars over the years, this perspective is used as a theoretical lens in this study to better interpret the results [33]. 

## 2. Literature Review and Hypotheses Development

Model overview: We conceptualize paradoxical leadership and employees’ voice behavior in our theoretical framework that team size influences the mediating effect of self-efficacy and psychological safety in the relationship between paradoxical leadership and voice behavior (see Figure 1). We adopt dual conceptualization of voice behavior, i.e., promotive voice behavior and prohibitive voice behavior, consistent with the works of Liang, Farh, and Farh [29] and Zhang, Huai, and Xie [34]. In line with leader-member exchange theory, we also argue that paradoxical leadership traits focus on establishing high quality exchange of dialogues between members of organization that leads employees to trust their leaders, which in turn enhances their voice behavior. Nonetheless, the interrelationship between self-efficacy and psychological safety is not considered under the scope of this study because the influence of one variable on the other is moderated by several factors [35]. In order to keep focus on main variables, the interrelation between mediating variables is avoided in the absence of direct link.

### 2.1. Paradoxical Leadership, and Voice Behavior

Zhang et al. [18] (p. 539) defined paradoxical leadership as “seemingly competing, yet interrelated, behaviors to meet structural and follower demands simultaneously and over time”. The notion behind the concept of paradoxical leadership is that “both-and” methods are adopted by the leader that behaviorally integrate and accept opposite demands simultaneously in order to gain from the intent behind the paradox [17,18]. Accordingly, the authors also proposed that paradoxical leaders encourage all team members to voice their concerns, ideas, and opinions, while at the same time, respecting each member’s viewpoint. 

The paradoxical leaders simultaneously offer guidance and instructions to attain innovative goals and encourage information sharing across teams. Considering such managerial practices, paradoxical leaders tackle integration and differentiation parameters for promoting innovation in diverse teams [3]. Zhang et al. [18] categorized paradoxical leadership into five behavioral dimensions, namely, maintaining decision control, while allowing autonomy; enforcing work requirements, while allowing flexibility; treating subordinates uniformly, while allowing individualization; maintaining both distance and closeness; and combining self-centeredness with other-centeredness. Lewis [36] found that paradoxical leadership carries out certain key roles in organizations by bringing both stability and flexibility that helps firms to manage in a changing external environment. These competing value dimensions of paradoxical leadership can be examined to understand how varied leadership paradoxes may encourage or restrict voice expression among employees.

According to Van Dyne, Ang, and Botero [37], voice behavior means the expression of employees about issues and situations pertaining to work environment. Such situations or issues include, but are not limited to, organizational functioning, task practices, and worries pertaining to business operations [28]. LePine and Van Dyne [35] stated that voice behavior is the behavior that is not obvious but expresses constructive viewpoints that are intended toward the improvement of an organization. Similarly, Van Dyne, Ang, and Botero [36] described voice behavior as a determined notion of one’s opinions and ideas towards potential progress. Morrison [38] however stated that voice behavior is associated with several constructs such as upward communication, whistle-blowing, or issue selling that do not directly suggest what voice behavior means. For example, voice behavior was linked to a response to dissatisfaction at work and was associated with something that intends to upgrade the working conditions [39]. Nonetheless, voice behavior can result into numerous positive outcomes such as prevention of business crises [40], organizational learning [41], and refining work processes [42]. Building on the arguments of voice scholars, the notion of voice behavior can be considered as constructive in intent, and therefore we theorize on the same to yield value.

Liang, Farh, and Farh [29] categorized voice behavior into prohibitive and promotive voice. The prohibitive voice essentially focuses on disposing harmful conditions while the promotive voice emphasizes on enhancing work practices or sharing suggestions. Lin and Johnson [43] found that prevention focus is positively related to prohibitive voice behavior, while the promotive voice behavior is positively linked to promotion focus. In addition to this, it was also found that depletion was positively linked to the prohibitive voice while it was negatively related to the promotive voice, which suggested the dynamic behavioral model of voice. Liang, Farh, and Farh [29] stated that the main focus area of existing studies around voice behavior has been the "promotive" aspect rather than “prohibitive” aspect, as the latter involves potential risks which are often misperceived by the colleagues. Burris [44] stated that as the prohibitive voice includes some practices and suggestions that cease some tasks or organizational processes, it ultimately questions the status quo of senior leaders. However, such voices about processes that inherit potential risks can prove to be a blessing in disguise, as that could save organization from big losses. The voice behavior concept was widened by Van Dyne et al. [37] who explained that both aspects of voice behavior are beneficial and productive for the firm. Hassan, Hassan, and Batool [45] concluded that both prohibitive and promotive voices are the proactive behavior of employees. While the prohibitive voice exhibits protective traits that highlight risky practices at workplace, the promotive voice on the other hand is suggestion oriented and thus constructive in nature. By studying both aspects of voice behavior, we create a thorough understanding of how paradoxical leadership is also related to prohibitive voice and not only promotive voice [38].

The organizations need to cultivate paradoxical leadership as they unlock employees’ potential by harnessing their talent, guiding them, and listening to their concerns. The employees feel risky when involved in groups or when they work under pressure, because speaking up in such circumstances may question the status quo of those who were responsible for the respective situation or who devised it [3]. The authors also concluded that when the voice behavior of a group is concerned, paradoxical leadership has a positive influence as they facilitate information sharing and focusing more on team success rather than individual success. This leads to the synergy in group tasks, which has catalytic effects on team learning and innovation. Considering the significance of concerns and worries of different team members regarding organisational functioning, the understanding of reflection of efforts of paradoxical leaders on employees’ voice behavior is pivotal.

The supervisors or leaders are the first targets in case of a voice process, so it is important that they must address this situation positively. Additionally, leaders have the power to decide punishments and rewards, and this authority over subordinates’ job tasks, promotions and pay makes the leaders’ actions highly accountable for their behavior [46]. Paradoxical leaders in such situations exhibit both controlled and freedom in behavior, as they are interested as well as willing to address the subordinates’ voice behavior and encourage them to speak up. Thus, when leaders send signals that they are interested in and willing to act on subordinates’ voice, subordinates are motivated to speak up [3]. Edmondson [47] also stated that subordinates were found to be more willing and contributing to teams when they encounter positive encouragement from their leaders in response to their speaking up. This research is noteworthy in that it investigates on how competing values in paradoxical leadership enhance the psychological sense of safety in speaking up.

Authors have confirmed that voice behavior is significant in facilitating innovation processes when employees bring up new ideas [11,48]. As novel ideas seem less promising in initial stages, they have high chances of being turned down and this is called the innovation paradox [49]. Additionally, the motivated information theory postulates that when teams receive non-redundant information and knowledge, the efficiency of executing tasks increases with collective efforts. The paradoxical leaders stimulate team perspective and balance the team innovative performance with team diversity, respond to the diverse demands of employees, and ensure that the social cues and risk factors do not restrict voice behavior of employees [50]. Thus, it can be inferred that paradoxical leadership affects employee voice behavior positively. Accordingly, we hypothesized the following-

*H1a*:
*There is significant influence of paradoxical leadership on employees’ promotive voice behavior*


*H1b*:
*There is significant influence of paradoxical leadership on employees’ prohibitive voice behavior*


### 2.2. Mediating Role of Self-Efficacy

According to Mensah and Lebbaeus [20], self-efficacy can be defined as the belief of an individual about his or her capability of executing a task. The authors have confirmed that higher the self-efficacy, the higher will be their confidence in the ability to make decisions and complete a task successfully [20]. It means that in difficult conditions, people with higher level of self-efficacy are more likely to deal with the situation properly than the ones with lower degree of self-efficacy, as the latter will try with reduced efforts or will eventually give up. The significance of self-efficacy arises when management of the organizations assigns numerous tasks to their respective employees and entrusts them to perform those tasks perfectly, which is reflected in their work-related behaviors as well [14]. Thus, it is essential for the organization’s management to be aware about the employee’s sense of efficacy as it impacts his or her job performance and ultimately affects the voice behavior.

Bandura [51] initially developed the hypothesis that self-efficacy influences achievement, persistence, efforts, and activities. This hypothesis served as a theoretical base to analyze traits, both theoretically and practically. For example, Schwarzer and Jerusalem [52] developed a generalized self-efficacy scale. Similarly, Luthans and Peterson [31] conducted an empirical investigation, which revealed that the self-efficacy of a manager was a partial mediator between the manager’s effectiveness and his/her subordinates’ degree of work engagement. The results of the study suggested that both the manager’s self-efficacy and subordinate engagement are key precursors that better jointly forecast a positive correlation with manager’s effectiveness rather than individual factors. This indirectly emphasizes the relationship between the role of a leader and self-efficacy. Similarly, drawing upon the theoretical base of work engagement events and social cognitive theory, the study of Yakin and Erdil [53] found that work engagement and self-efficacy influence job satisfaction. Hence, it is suggested that leadership positively influences employees’ self-efficacy and in turn, the work outcomes.

Heuven et al. [54] concluded that people who have strong beliefs about their ability to perform tasks successfully set difficult objectives for themselves, try harder, invest more, and deal with situations better than their counterparts. They also make better use of their resources and skills to deliver output to the challenging tasks assigned to them. As such, all these attributes indicate that people with high self-efficacy exhibit more inclination for completing their tasks perfectly and signal for promotive voice behavior. However, Judge et al. [55] stated that task or job complexity is one of the most obvious moderators of self-efficacy predictiveness. In this context, Kanfer and Ackerman [56] found that distal traits rather than self-regulatory/self-efficacy are more relevant in predicting performance in case of complex tasks. This argument of self-efficacy was also supported by Stajkovic and Luthans [57] and Chen, Casper, and Cortina [58] who suggested that self-efficacy might not be directly related to performance and voice behavior. In addition to this, Judge et al. [55] asserted that self-efficacy is bounded by the involvement of discrete opinions and is valid in cases where conditions are theoretically similar to the practical ones, which may not always hold true. On the basis of the above discussion, self-efficacy is positioned in a mediating role in our study.

Xie et al. [21] stated that as voice behavior requires a person to have relevant abilities and expertise to analyze potential problems and to provide solutions to them, voice self-efficacy is considered the key to it and can be developed by frequent experiences of speaking up in organizations. Additionally, employees having higher self-efficacy believe that they make a positive contribution to the teams and organizations in comparison to those having low self-efficacy, as they can better express their ideas and overcome fears through their voice behavior [59]. Tierney and Farmer [60] discussed the creative notion of self-efficacy which advocates an individual’s beliefs pertaining to the ability and skills to generate creative outcomes, and this has been found to forecast employee creativity and to moderate the impact of several factors on creativity [23,61,62]. Therefore, the existing literature suggests a clear link between employees’ self-efficacy and their willingness to indulge into decision making and voice behavior.

Zhang et al. [18] suggested that paradoxical leaders act as role models for individuals, explaining them how to deal with situations and complex tasks, as at workplaces, they practically show how to deal with situations constructively, how to derive the meaning, how to handle a situation and how to form a solution. Bandura [63] highlighted that such vicarious learning thus determines the formation of self-efficacy and explains the significance of paradoxical leadership. The individuals process cues and information from their surrounding environment to generate efficacy judgements [51,64]. To help with judgments in work environment, a paradoxical leader is the one who guides them about the goals, controls the flow of information, resources, and decides punishments and rewards and therefore, a leader is a key influencer in forming self-efficacy [65,66]. Additionally, it was also found that by targeting role modeling behaviors, the leaders can increase the self-efficacy of their followers [22,23] and by creating such supportive environment for employees to balance tensions in creativity, the paradoxical leadership behavior eventually strengthens the self-efficacy of employees [24]. The results of these studies have proven that paradoxical leadership influences self-efficacy and plays an important role in forming an employee’s voice behavior. Accordingly, the following are hypothesized-

*H2a*:
*There is a significant mediating effect of self-efficacy in the relationship between paradoxical leadership and employees’ promotive voice behavior*


*H2b*:
*There is a significant mediating effect of self-efficacy in the relationship between paradoxical leadership and employees’ prohibitive voice behavior*


### 2.3. Mediating Role of Psychological Safety

According to Edmondson [41] (p. 354), psychological safety is defined as the perception of members in which they feel comfortable in being themselves, in interpersonal context. A psychologically safe environment is referred to the one in which individual members are to express themselves freely, such as their self-doubts, concerns, and their needs for learning so that they could work constructively [14] (p. 708). Walumbwa and Schaubroeck [26] also said that psychological safety comprises of experiencing and perceiving an increased degree of interpersonal trust. It is also characterized by a work climate wherein people have mutual respect and are comfortable in working with each other. The rationale behind psychological safety is realized when members believe that failures are a part of learning and speaking up as well as inquiring about anything and is not subject to any form of repercussion from their leaders or other members.

Yang et al. [67] stated that psychological safety is an important boundary factor that facilitates creativity at workplace and gives deeper insights about paradoxical leadership behavior. Numerous studies suggest that job outcomes are subject to positively thriving at a workplace and the successful employees who thrive at workplaces more are able to identify or create new opportunities and processes for the problems that eventually play a role in enhancing their creativity [68,69,70]. However, Dewett [71] argued that such growth of creativity and ideas at workplaces is subject to substantial form of risk and may even face hindrance in the form of work safety, norms, balance of power, relationships or even failures [72]. Such risks and vulnerabilities function to restrain the behavioral outcomes and growth of individuals as well as of organizations and therefore, this study draws on psychological safety as an essential parameter [73,74].

As such, a psychologically safe environment is supposed to enhance an increased degree of employees’ well-being rather than an unsafe one, because in such an environment, people have more voice behavior, and feel free from any external constraints, bias or control [69,70]. Employees who perceive higher levels of psychological safety therefore place more trust in their leader, engage in voice behavior, take risks and have parallel thinking that all together boosts their whole well-being [43]. Contrarily, employees exhibit low voice behavior and feel reluctant to get into task-related processes such as asking for resources, questioning, reporting problems, seeking feedback, offering ideas and solutions in an environment that offers low level of psychological safety [66]. Further, uncertainty and fear are also present in such environment and often distracts employees and thus reduces productivity. Farmer, Tierney, and Kung-Mcintyre [75] also stated that subordinates in a low-psychologically safe environment perceive no enough support from their leaders about their well-being and growth.

To behave creatively, explore novel ideas, and willingness to think innovatively requires an umbrella of psychological safety wherein the risks of such process become shielded [66]. Baer and Frese [76] also found evidence about the positive link between an environment of psychological safety and innovativeness and concluded that the former is essential for creativity in work processes. This is because creative inventions and novel ideas look similar to unrealistic and ridiculous at first glance and in the initial stages the value of such ideas contain ambiguities, risks of committing mistakes and uncertainties, which may take some time to bear fruit. In an environment of low level of psychological safety, negative personal outcomes (such as being sanctioned at times, perceived as foolish, or decrease in respect) could emerge due to the absence of creative work. As a result, individuals in such an environment feel afraid and reluctant to open up and hence reflect low voice behavior.

The voice behavior involves several kinds of risks [29,33]. For example, employees constantly think that their voice behavior could be misunderstood or reprised by their supervisors, eventually deteriorating their relationship [7]. In such scenarios, psychological safety acts as a key tool to evaluate such risks and dampens any negative aspects associated with increasing voice behavior [3]. Paradoxical leaders understand the importance of psychological safety and thus emphasize more on fostering and sustaining an open environment. Edmondson [41] found that team learning behaviors and creativity is positively related to psychological safety [77]. In another study about interdisciplinary action teams that examined learning, Edmondson [41] found that effective team leaders could promote innovation and facilitate learning by fostering an environment of psychological safety. In support of these arguments, we assume that psychological safety works as a mediating function between leadership and voice behavior.

Several studies have found evidences that psychological safety mediates the link between leader-employee exchange and voice behavior [32]. In another related research about the results of psychological safety, Newman, Donohue, and Eva [72] concluded that an increase in voice behavior is found in scenarios where paradoxical leadership increased employees’ psychological safety, both within teams and individually [3,26]. Further, social information processing theory states that the attitudes and behaviors of people are affected by the interaction with their social environment [78]. At work, employees inculcate several forms of information that they use to evaluate voice behavior risks and decision making. Paradoxical leadership traits focus on establishing high quality exchange of dialogues between members of organization that leads employees to trust their leaders. This opens the barriers about safety concerns and enhances voice behavior [79,80]. In all mentioned studies, authors have considered psychological safety as a unidimensional construct and mediating variable. Accordingly, following are hypothesized:

*H3a*:
*There is a significant mediating effect of psychological safety in the relationship between paradoxical leadership and promotive voice behavior of the employees.*


*H3b*:
*There is a significant mediating effect of psychological safety in the relationship between paradoxical leadership and prohibitive voice behavior of the employees.*


### 2.4. Moderating Role of Team Size

The motivated information processing theory postulates that in order to increase the depth and magnitude of innovative information processing, groups in an organization should not only ensure that information should be integrated and shared as a collective effort, but they should also function in methods that embrace distinct viewpoints and perspectives. This becomes increasingly difficult when the size of the groups is large [81,82]. Though paradoxical leaders respect the views of team members and motivate them to respect other members’ differing opinions, it becomes challenging when the size of the team grows [18]. Hooper, Kaplan, and Boone [83] also stated that leadership is important in scenarios where the group size increases and coordination among the individuals fails, owing to free-riding behaviors. In such cases, the presence of a leader increases the cooperation among team members as they prefer to work under his or her supervision than to work in an unsupervised group. Li, She, and Yang [3] further stated that paradoxical leadership is essential for teams that have varying sizes and diversity to promote and fuel innovation by overcoming the integrating-differentiating paradox in teams. These findings call for an explanation as to how paradoxical leaders face challenges in balancing competing behavioral dimensions of paradoxical leadership with increase in group size, which might influence employees’ willingness to speak up.

The paradoxical leader treats his subordinates uniformly while granting individualization [18]. This means that the leader assigns his subordinates with similar status and rights without showing any favoritism [35]. The leader in small teams will be able to recognize the team members in a better way by having proper leader member exchange communication [25]. The subordinates in such scenarios have a close relationship with their supervisor and could exhibit a high level of voice behavior. As such, it will be expected from the team members to engage in more self-driven behavior and thus have higher self-efficacy levels. The opportunities in such scenarios will also be more for the team members to grow, innovate, improvise, and learn [3].

If the size of a team is small, a paradoxical leader will have a better view and understanding about the performance and behavior of his or her subordinates. The small size of the team means that the roles and responsibilities of the individual in that team will be more as compared to the case when the team size is high. On the other hand, if the team size is bigger, the paradoxical leader will not be able to interact and communicate with all the subordinates, and thus exhibit more controlled behavior. Only selected subordinates in big teams will have a close connection with the team leader, and thus will have higher voice behavior than others in the team. In such cases, the other team members will have uniformity, which could depersonalize them and discard their unique individual skills [84]. Similarly, since the team size is big, not all the team members have enough motivation to indulge in innovative and creative work, which often makes the self-efficacy level low [85].

Detert and Trevino [86] stated that when the team members exhibit higher level of voice behavior, they could voice problems and alternative ideas that could improve the organizational and team effectiveness. However, since big teams have increased hierarchies that often impede voice behavior of members with lower status, the role of leaders in motivating subordinates to speak up is highlighted. Weiss et al. [27] also found that employees and subordinates hesitate frequently to voice concerns and ideas across different level of hierarchies. The authors also found that implicit leaders such as paradoxical leaders emphasize more on “We, Us and Our” language and thus, encourage employees to speak up about their concerns and alternative ideas. 

Li, She, and Yang [3] stated that in situations where team perspective is not strong and team diversity is prominent, members of the team in such cases devote more time and efforts in working on their own ideas rather than seeking comments from others in the team, thereby lacking collectivity and self-efficacy. Hoever et al. [87] found empirical evidence that in multidisciplinary team, putting team perspective in low regards shows resistance while accepting perspectives and information from other team members. Disregarding the value of ideas which others could see, increases the risk of committing mistakes, leads to a decreased overall efficiency of the team and correlates negatively to innovation [88]. 

Brown [89] stated that team members consider asking for help, seeking feedback and admitting error as a threat to their face and thus, are hesitant to raise their voice behavior even when it is beneficial to the team [90]. This situation becomes more frequent when the size of the team increases. Edmondson [41] conducted a study taking 51 teams of a manufacturing company as a sample space and found that when team structure grows and becomes more complex, learning behavior mediates between team performance and team psychological safety. Therefore, team members have higher psychological safety levels in small teams with increased perceived consideration from leaders. This in turn increases the voice behavior of the members. Thus, this study assumes that small teams exhibit a higher level of psychological safety and self-efficacy, which increases the voice behavior of team members and decreases as the team size increases. Accordingly, the following are hypothesized:

*H4a*:
*There is significant moderating effect of team size on the mediating effect of self-efficacy in the relationship between paradoxical leadership and employees’ promotive voice behavior.*


*H4b*:
*There is significant moderating effect of team size on the mediating effect of self-efficacy in the relationship between paradoxical leadership and employees’ prohibitive voice behavior.*


*H4c*:
*There is significant moderating effect of team size on the mediating effect of psychological safety in the relationship between paradoxical leadership and employees’ promotive voice behavior.*


*H4d*:
*There is significant moderating effect of team size on the mediating effect of psychological safety in the relationship between paradoxical leadership and employees’ prohibitive voice behavior.*


## 3. Research Approach

### 3.1. Sample and Data Collection

This study has been approved by the Research Ethics Committee of the university before we collected the data, and the study did not violate any legal regulations or common ethical guidelines. We distributed and administered a self-report questionnaire to 360 supervisors and subordinates in South-Western China, whom were paid a reasonable time of 30 minutes to fill the survey questionnaire. After removing the responses for missing data, the data concerning 155 subordinates and 96 supervisors was run through analysis. The survey was directly administered, and data was kept under safe personal custody of the researcher to ensure confidentiality. Additionally, the order of the measures was randomized in order to reduce the possibility of respondents providing similar responses. The employees were asked to respond to the paradoxical leadership behavior of their supervisors, their self-efficacy and perceived psychological safety. On the other hand, the supervisors were asked to respond on the promotive and prohibitive voice behavior of their employees. All the respondents were informed about the research purpose. 

Nearly half (51.6%) of the subordinate respondents were found to be younger, i.e., under the age of 40 years, while only 9% were found to be older, i.e., more than 50 years. Around 57.6% of the subordinate participants were males and nearly 54.2% were married. In relation to work experience, almost half of the subordinate participants had work experience of less than 3 years. For supervisor respondents, only 11.6% of the participants were younger and underage, i.e., of 30 years, while most of them (32.9%) fell into the age group of 31 to 40 years. The supervisors also possessed lesser work experience, with nearly half of them having experience of less than 3 years. 

### 3.2. Measures

Paradoxical leadership: To measure paradoxical leadership, 12 items for 5 factors were adapted from the study of Zhang et al. [18]. It should be noted that the original scale of Zhang et al. [18] comprised of over 20 items for 5 dimensions. However, during a pilot study to validate the instrument, it was identified that it was very time consuming for employees/supervisors to take out much time from their busy schedule to respond to 4 or 5 items per dimension of a single variable. Accordingly, to support the voluntary and objective response behavior of target respondents, the item scale was shortened to at least 2 items per dimension in the main study. However, it was ensured that only the items selected for measurement provided a comprehensive view of the five dimensions of paradoxical leadership. The selection of shortened scale items was based on review of existing literature, as to what items are widely supported by authors to define or measure paradoxical leadership. All the items were constructed on a five-point Likert scale, ranging from “strongly agree” to “strongly disagree”. For each of the five dimensions of paradoxical leaders, at least two items were designed, with a total of 12 items. The measure across these five dimensions was selected over others because these provided a comprehensive view of paradoxical leadership based on the literature review. An example item was “maintains an unbiased relationship with all individuals in the team”. 

Psychological safety: We tested the psychological safety construct at the level of work-unit analysis as proposed by the classical work of Edmondson [41]. We instructed to mention their agreement level to three items using the five-point scale, ranging from “strongly agree” to “strongly disagree”. The sample items include “my honest feedback and feelings are advocated at workplace” and “sometimes I fear that voicing out my opinions to others might backfire”.Self-efficacy: We asked the subordinates to rate their level of agreement to a set of 10 items. The scale of 10 items was borrowed from the work of Schwarzer and Jerusalem [52]. The scale of self-efficacy was created from generalized sense of perceived self-efficacy with consideration of coping with stressful daily hassles and life events [20]. The items were scaled on a five-point Likert scale ranging from “strongly agree” to “strongly disagree”. The sample items were “I feel like I can figure out multiple options when asked about a problem” and “I keep myself engaged in some work or another, even if not assigned any official task”.Voice behavior: There was dual conceptualization of voice behavior in this research, promotive and prohibitive voice behavior as theorized by Liang, Farh, and Farh [29]. The constructs were measured using three item scales for each of the promotive and prohibitive voice behaviors, as borrowed from the study of Liang, Farh, and Farh [29]. The supervisors were asked to respond on their level of agreement on these items considering their subordinates’ voice behavior on a five-point Likert scale, ranging from “strongly agree” to “strongly disagree”. An example of item for promotive voice behavior was “recommends idea that improve processes and workflow which proves to be beneficial for the team”. An example of item for prohibitive voice behavior was “honestly voice opinions, issues and feedback to supervisor(s) and other colleagues, even though different opinions exist”.Team size: The review of existing literature suggested that only a few authors have considered the effects of group or team size in the voice and leadership research [9,27]. Therefore, this research extended the literature by considering the moderating effects of team size on the role of self-efficacy and psychological safety on leadership-voice behavior linkage. Since the previous studies do not provide established scales to measure team size effect, we developed the items based on the findings and theory of authors on team characteristics researches (for example, [25]). We carefully developed the scale items in a manner that no two items overlap each other. The items were created on a Likert scale of five-point, ranging from “strongly agree” to “strongly disagree”. An example item was “in a large team, people often engage in self-driven behavior”.Control variables: We controlled for the effect of both subordinates’ and supervisors’ demographic characteristics, which may exert an influence on their behavior. The demographic characteristics included age, gender, and work experience (work tenure).

## 4. Data Analysis

### 4.1. Measure Validation

In order to test the reliability and validity of measures used in questionnaire, we utilized SPSS software v23.0 (IBM, Almaden, CA, USA)to conduct the factor analysis. After deleting a few items with a low factor loading, the model showed a good fit (Table 2). The KMO variable was obtained to be 0.635, which is acceptable and indicated the adequacy of each of the variables and intercorrelations in the set of variables in the given dataset. The results of the Bartlett’s test of sphericity suggest that the p value is 0.000, which means that the factor analysis results are statistically significant (Table 3).

Using SPSS v23.0, we conducted Herman’s single factor test on the survey data. The total variance explained by a single factor came out to 5.467%, which is far less than the totality by 50%. The results indicate that the given dataset does not suffer from the common method bias. Therefore, the variations in the responses of the survey respondents are not caused by the instrument, but the actual predispositions of the survey participants.

### 4.2. Descriptive Statistics

With the help of SPSS v23.0, descriptive statistics and correlation tests were applied, the results of which are presented in Table 4. The results of descriptive statistics reflect average responses of the respondents around the five-point Likert scale. The leaders’ responses on employees’ voice behavior show that employees were found to be more engaged in promotive voice behavior (mean value 1.85) as compared to prohibitive voice behavior (mean value 2.57). On the other hand, employees’ average agreement level on their self-efficacy (mean value 2.01) was stronger than on psychological safety (mean value 2.58). Additionally, the standard deviation for all the factors was below 1, suggesting that there was no significant deviation of the mean values from the observed responses.

The results of correlation tests revealed that there is significant and positive correlation between paradoxical leadership and a) promotive voice behavior (*r* = 0.162; *p* < 0.05), b) psychological safety (*r* = 0.076; *p* < 0.05) and c) self-efficacy (*r* = 0.024; *p* < 0.05). While there is significant and negative correlation between paradoxical leadership and prohibitive voice behavior (*r* = −0.372; *p* < 0.01). These results were consistent with the alternative hypotheses 1a and 1b developed in the literature review section.

### 4.3. Model Estimation

Using SPSS v23.0, regression analysis was applied on the dataset to test the proposed hypotheses. The main effects model, as depicted in Table 5, revealed that paradoxical leadership is negatively related to both self-efficacy (*b* = −0.004, *p* < 0.01) and psychological safety (*b* = −0.287, *p* < 0.01). 

To test the mediation effects, our conceptual model hypothesized the mediating effect of self-efficacy and psychological safety. The indirect effects of paradoxical leadership on promotive voice behavior and prohibitive voice behavior through self-efficacy and psychological safety were estimated by bootstrapping 5000 samples at 95% confidence interval (CI). The mediating effects were estimated using “Process v3.4 by Andrew F. Hayes” in SPSS v23.0 with the help of Model no. 7, which considers mediating and moderating effects. 

The paradoxical leadership was not found to have significant direct effect on any of the two dependent variables, promotive and prohibitive voice behavior. Additionally, self-efficacy does not have a significant indirect effect on the relationship between paradoxical leadership and promotive voice behavior (*b* = 0.0700, CI (−0.0199, 0.0573)) and on the relationship between paradoxical leadership and prohibitive voice behavior (*b* = 0.0102, CI (−0.0793, 0.0491)), which is not in support of H2a and H2b. 

On the other hand, psychological safety has a significant indirect effect on the relationship between paradoxical leadership (*b* = 0.0285, CI (0.0274, 0.0491)) and promotive voice behavior and an insignificant indirect effect on the relationship between paradoxical leadership and prohibitive voice behavior (*b* = 0.0179, CI (−0.0647, 0.0914]), which is in support of H3a, but not H3b.

We tested the interaction effects between mediating and moderating variables through ‘Process v3.4 by Andrew F. Hayes’ in SPSS v23.0 with the help of Model no. 7. The results revealed that team size has no significant moderating effect on mediating role of self-efficacy in the relationship between paradoxical leadership and promotive voice behavior (*b* = 0.0700, CI (−0.0199, 0.0573)) and paradoxical leadership and prohibitive voice behavior (*b* = 0.0003, CI (−0.0694, 0.0313)), which is inconsistent with alternative hypotheses H4a and H4b. In relation to H4c, the effect of psychological safety on employees’ promotive voice behavior increases as the team size increases (*b* = 0.0163, *p* < 0.05) which accepts the alternative hypothesis. It should be noted that the effect of psychological safety on promotive voice behavior is stronger at low levels of team size (*b* = 0.0285, *p* < 0.05) as compared to high levels of team size (*b* = 0.0122, *p* < 0.05). While there is no significant moderating effect on the mediating role of psychological safety in the relationship between paradoxical leadership and prohibitive voice behavior (*b* = 0.0102, CI (−0.0793, 0.0491)), which rejects H4d. See Figure 2.

## 5. Discussion

### 5.1. Theoretical Implications

Although prior research has investigated the relationship between leadership and voice behavior [3,31], the literature on impact of paradoxical leadership and voice behavior is still undeveloped. Drawing on leader-member exchange theory, we extend the scope of existing literature by identifying the impact of paradoxical leadership on employees’ voice behavior. Our research contributes to the literature in three ways. The major contribution is to identify if the paradoxical leaders evoke voice behavior of employees through an exchange process. We empirically found that there exists an intricate link between paradoxical leadership and voice behavior. When leaders adopt paradoxical behavior, employees are more likely to engage into promotive voice behavior; however, employees’ prohibitive voice behavior is reduced when their leaders adopt paradoxes in their leadership behavior. These findings reflect that with appropriate leader–member exchange, employees’ psychological safety is increased, which in turn promotes voice behavior of employees. This is because leader–member exchange theory provides that leaders’ close relationships with followers enhances positive work attitude, which might encourage their voice behavior. However, the need to maintain distance arises to protect followers’ charismatic attributions towards leaders, which again causes followers to be in a hierarchical relationship and they tend to restrict voicing out on issues and problems, i.e., prohibitive voice behavior [18]. Therefore, we discovered a leader–member exchange mechanism that inherently originates from the intention of paradoxical leadership [3,26,79,80]. 

Second, we found that psychological safety has significant indirect effect on the relationship between paradoxical leadership and promotive voice behavior and no significant mediating effect on the relationship between paradoxical leadership and prohibitive voice behavior. Our results are consistent with the evidences of prior research in leadership–voice behavior linkage that psychological safety mediates the link between leader–employee exchange and voice behavior [3,26,36]. On the other hand, self-efficacy has no significant mediation effect on the relationship of paradoxical leadership with promotive voice behavior and prohibitive voice behavior. The prior research has widely considered the mediation effect of psychological safety in leadership and voice literature [3,26,27], but self-efficacy in this area has been poorly studied. Nonetheless, our research found no significant mediating effect of self-efficacy in this context.

Third, we tested the moderating effect of team size on the mediating role of self-efficacy and psychological safety on the relationship between paradoxical leadership and employee voice behavior. We found that team size has no interaction effects with self-efficacy on employees’ promotive as well as prohibitive voice behavior. On the other hand, team size has significant interaction effects with psychological safety on promotive voice behavior, while insignificant interaction effects on prohibitive voice behavior. Specifically, the effect of psychological safety on promotive voice behavior is stronger at low levels of team size as compared to higher levels of team size. Our results thus make a significant contribution to the team characteristics literature by extending the focus of team size as the controlling factor to moderating effects and suggesting that with smaller team size, when the employees consider themselves as psychologically safe, they are more likely to engage into promotive voice behavior under paradoxical leadership [9,27].

We articulate a viewpoint that paradoxical leaders need to trigger higher levels of exchange of dialogues within their team in order to encourage voice behavior of employees. In this manner, our research extends the leadership and organization behavior research on the leader–member exchange effect. The leader-member exchange effect can be applied in the context of proactive behavior of employees including voice behavior.

### 5.2. Practical Implications

Our empirical results offer meaningful implications for the managers in practice on how paradoxical leadership behavior can be used to stimulate the voice behavior of employees. Our findings revealed that when a safe environment is provided to employees, paradoxes in leadership behavior can encourage more promotive voice behavior among the employees. The managers need to consider that a psychologically safe environment provided to the employees results in their proactive behavior, creativity, and voice out behavior because they place greater trust in their leaders [47,67,68,69,70,91]. Additionally, managers need to understand that in small teams, subordinates perceive a close relationship with their supervisors and therefore, supervisors try to improvise and learn [3,25]. Therefore, in addition to self-concept and a psychologically safe environment, managers can also engage into strong leader–member exchange effect and thereby encourage employees to voice out. When the subordinates think that they have close relationship with their supervisor in the small teams, they tend to speak up.

The prior research has suggested that team members consider asking for help, seeking feedback, and admitting error as a threat to their face and thus, are hesitant to raise their voice behavior even when it is beneficial to the team [70,89,90]. Our study suggested that paradoxical leaders can offer reasonable level of flexibility and autonomy to employees, so that they speak up on problems and suggestions to solve them, while maintaining the final decision-making power. Therefore, the paradoxical leaders need to exercise their completing leadership behaviors in a manner so as to facilitate a direct and strong exchange of dialogues with their team members. For this purpose, effective communication strategies such as two-way feedback mechanism should be promoted so that followers better interpret the expectations of the paradoxical leaders.

One important indicator of system-level change concerns the degree of authority, hierarchy, and respect attributed to individuals in a social system that restricts the desired level of psychological safety and this results in employees underperforming in their tasks in the workplace environment [92,93]. In this context, self-efficacy is one important component of an individual’s self-structure and a key psychological moderator that affects the relationship between occupational burnout and stress in people. Perceived self-efficacy thus turns into a psychological variable that motivates people and is responsible for the effectiveness of various workplace activities [63]. Moreover, as employee safety concerns also involve the physical safety of employees, governments and corporations should introduce regulations that concern safety protocols at work or procedures that involve occupational health and safety. Additionally, organizations also need to create psychological safe environment to facilitate learning behaviors and develop team cohesion. To create an open environment with increased level of psychological safety, managers should encourage the admission of mistakes by themselves in front of other co-workers [59]. By adopting and implementing such practical solutions for the real-world problems, paradoxical leaders could establish better leader-member exchange and promote voice behavior.

### 5.3. Limitations and Future Research

The reliability and validity of the results of our empirical study and its theoretical and managerial contributions should be evaluated and understood in light of several limitations of this study. First, we examined longitudinal data in this study where subordinates and supervisors were surveyed at a single point of time. However, due to use of longitudinal data, we could not preclude casual inferences. In future, our research can be replicated with time-lagged data to obtain a better understanding of reciprocal casualty in the voice behavior of employees in China. 

Second, since we collected data from the subordinates and supervisors of different Chinese organizations, our results may not hold validity in other cultures as well. The competing behavior of paradoxical leaders may result in meaningful work outcomes when there is strong leader–member exchange in addition to a safe environment. However, these findings are more generalizable in the context of the Chinese businesses. Authors have established that the leaders in the Chinese culture follow paternalism and social orientation [33,91]. The social orientation and paternalism among the Chinese leaders help them adopt a better communication process in their leadership. On the other hand, Western cultures hold more individualistic characteristics [92]. It is therefore suggested that this study should be replicated in future by making a cross country examination of paradoxical leadership and voice behavior in Eastern and Western cultures.

Finally, our study was based on dual conceptualization of voice behavior in terms of promotive voice behavior and prohibitive voice behavior [29]. However, the voice research is developing recently authors have proposed an expanded version of voice behaviors including defensive voice behavior, constructive voice behavior, supportive voice behavior, etc. The future researchers are suggested to examine potential effects of paradoxical leadership with these voice behaviors. 

## 6. Conclusions

Drawing upon leader-member exchange theory, this study empirically tested the relationship between paradoxical leadership and employees’ voice behavior through indirect effects of self-efficacy and psychological safety and moderating effect of team size. Based on composite reliability and discriminant validity, the measurement scales which were used in this research were found to be reliable and valid. Additionally, the proposed model provided a good fit to the data. First, we demonstrate that the self-efficacy of employees and their psychological safety can be increased or decreased based on the paradoxical leadership behavior. Second, we find that paradoxical leadership has an indirect effect on employees’ promotive voice behavior through psychological safety. Third, the team size moderator was tapped into the model to show its effect on the mediating role of self-efficacy and psychological safety on the paradoxical leadership-voice behavior linkage.

## Figures and Tables

**Figure 1 ijerph-17-01162-f001:**
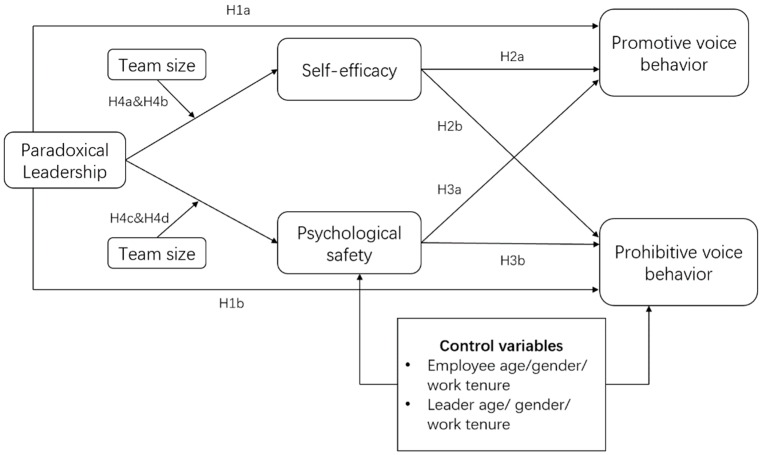
Framework.

**Figure 2 ijerph-17-01162-f002:**
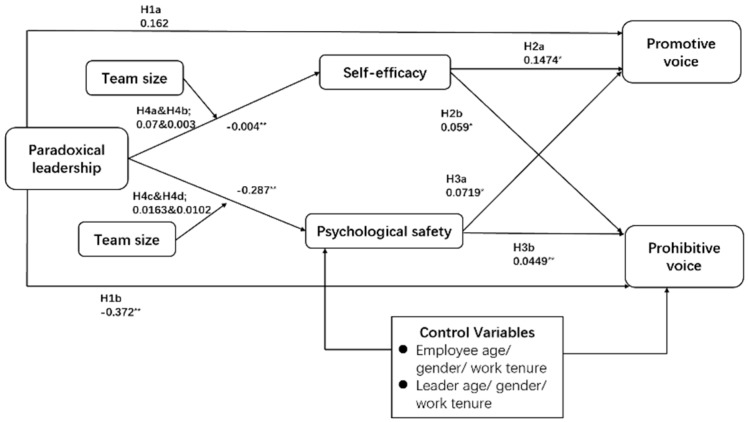
Results and coefficients.

**Table 1 ijerph-17-01162-t001:** Literature on leadership and voice behavior.

Study	Sample	Type of Leadership and its Link with Voice Behavior	Mediators/Moderators	Control Variables
Pearce et al. [2]	Qualitative thematic analysis of 78 formal interviews of leaders in a high growth retail firm	Situational leadership Formal leadership Shared leadership Paradoxical leadership Theme emerged under shared leadership: encourages others to voice	-	Social integration, performance, and well-being-related variables
Walumbwa and Schaubroeck [26]	A sample of 894 employees and their 222 immediate supervisors in a large financial institution in the South-western United States	Ethical Leadership Agreeableness Conscientiousness Neuroticism Influences followers’ voice behavior	Psychological safety	Leader’s span of control
Zhang et al. [18]	A sample of 516 subordinates and 76 supervisors HR managers from four companies located in Beijing in Mainland China	Paradoxical Leadership Combining other-centeredness with self-centeredness Maintaining both closeness and distance Enforcing both flexibility and work requirements Allowing individualization while maintaining uniformity Allowing autonomy with decision control	Leader–member exchange Status judgment	Supervisors’ and subordinates’ age Supervisors’ and subordinates’ gender Supervisor’s tenure Power distance Relational orientation
Detert and Burris [3]	A sample of 3153 crew members in 105 restaurants and 270 shift managers from Serve-Co restaurants	Transformational Leadership Managerial openness Improvement oriented voice	Subordinates’ perceived psychological safety	Demographic, personality and dispositional controls Tenure, job type, ethnicity, gender Employee attitudes, having ideasJob shift, proactive personality, hours worked per week
Weiss et al. [27]	A sample of 126 participants participated in the hospital’s one-day simulation training sessions	Inclusive Leadership Implicit leader language Explicit leader language Inclusive leader language promotes voice behavior	Authors did not test any mediating effects but suggested potential mediation effects Team identification Psychological safety	Total number of leaders’ utterances Team size
Frazier and Bowler [9]	A sample of 374 full-time employees across 54 work groups	Group perceptions of supervisor undermining	Group voice behavior Discussion with top management Supervisor ratings	Age Gender Group size Tenure
Gong, Zhou, and Chang [11]	A sample of 761 core knowledge employees, A sample of 148 CEOs, and 148 HR executives from 148 high-technology firms	Transformational Leadership (four elements of transformational leadership)	Creative self-efficacy	Age Gender Education level Employee rank Company tenure Insurance business related experience
Mensah and Lebbaeus [20]	A sample of 70 employees from service institutions, 50 employees from financial institutions and 80 employees from educational institutions	Self-efficacy	-	Employee age Tenure of employee
Venkataramani and Tangirala [28]	A sample of 184 bank employees nested within 42 work groups	Employees’ work-related centrality and voice behavior	Personal influence	Employee age Tenure of employee
Liang, Farh, and Farh [29]	A sample of 239 employees	Psychological safety effect on promotive and prohibitive voice behavior	Felt obligation for constructive change Organization-based self-esteem	
Luthans and Peterson [30]	A sample of 170 managers and 2720 employees	Self-efficacy	Employee engagement	Managers’ and subordinates’ proactiveness, attitudes and engagement
Duan et al. [31]	A sample of 146 leaders and 349 subordinates in the South-western China	Transformational leadership (four elements of transformational leadership)	Leader voice expectation Employee voice role perception Personal identification	Felt responsibility to change Psychological safety

**Table 2 ijerph-17-01162-t002:** Factor Loadings.

	Extraction
Paradoxical leadership	
Maintains an unbiased relationship with all in the team	0.611
Treats all team members uniformly but also respects their individual capabilities	0.718
Facilitates expressive and two-way communication uniformly with all, but also cares about the attitudes and needs in	0.708
By setting leadership examples for everyone, allows others to assume leadership role as well	0.704
Respects self-opinion and also respects other people’s opinions and values	0.733
Handles important tasks himself/herself but also delegates responsibilities to others	Deleted
Ask for opinions and feedbacks from team members before taking final decision	0.748
Maintains strict deadlines and timelines but also understands exceptional situations	0.640
Only briefs about the task and lets members do all the execution	Deleted
Keeps high expectations from team members about tasks but also advocates learning from mistakes	Deleted
Maintains a respectable boundary with subordinates	0.704
Respects formal behavior but keeps amiable relationship towards others	0.763
Psychological safety	
I am able to express my opinions, thoughts and suggestions freely at workplace	Deleted
My honest feedback and feelings are advocated at workplace	
Sometimes I fear that voicing out my opinions to others might backfire	0.797
Self-efficacy	
I am confident and focused on my work and goals	0.716
I know how to get my work done, even if someone raises any objection	Deleted
I feel that if unexpected situations come, I can deal with them comfortably using my resourcefulness, skills, and by staying calm	0.732
I feel like I can figure out multiple options when asked about a problem	0.798
I keep myself engaged in some work or another, even if not assigned any official task	0.787
Generally, I focus on my efforts to accomplish my tasks and objectives	0.686
I set my goals clear and keep myself motivated to achieve them	0.793
I consistently work to achieve my goals	0.779
I see mistakes as learning and target achieving small goals to achieve the ultimate success	0.806
Sometimes I fear that I might fail to execute my duties properly and miss my goals	0.581
Promotive voice behavior	
Proactively identify and suggest potential issues that affect the team	0.551
Recommends idea that improve processes and workflow, which proves to be beneficial for the team	0.702
Suggests new ways of doing things and projects	0.724
Prohibitive voice behavior	
Honestly voice opinions, issues and feedback to supervisor(s) and other colleagues, even if different opinions exist	0.663
Speak up about problems and things that appear inefficient to them, even if it affects their professional relationship with them	Deleted
Highlight and raise concerns about inefficiencies that exist at workplace to the higher authority	0.724
Team size	
When team size increases, I receive less individual consideration	0.717
In a large team, people often engage in self-driven behavior	0.646
In big teams, my leader keeps behavior controlled	Deleted

**Table 3 ijerph-17-01162-t003:** Results of factor analysis.

KMO and Bartlett’s Test		
Kaiser-Meyer-Olkin Measure of Sampling Adequacy.		0.635
Bartlett’s Test of Sphericity	Approximate Chi-Square	445.697
	df	351
	Significance	0.000

**Table 4 ijerph-17-01162-t004:** Descriptive statistics.

		PL	PS	SE	PMV	PHV	TS	Mean	SD
Paradoxical leadership	R	1						2.33	0.471
	Sig. (2-tailed)								
Psychological safety	R	0.076	1					2.58	0.836
	Sig. (2-tailed)	0.035							
Self-efficacy	R	0.024	0.033	1				2.01	0.197
	Sig. (2-tailed)	0.008	0.683						
Promotive voice behavior	R	0.162	−0.071	−0.075	1			1.85	0.523
	Sig. (2-tailed)	0.009	0.049	0.047					
Prohibitive voice behavior	R	−0.372	−0.098	−0.036	−0.195	1		2.57	0.722
	Sig. (2-tailed)	0.007	0.034	0.007	0.047				
Team size	R	0.006	0.109	0.040	−0.038	0.043	1	2.52	0.784
	Sig. (2-tailed)	0.009	0.179	0.620	0.710	0.007			

Note: PL = paradoxical leadership; PS = psychological safety; SE = self-efficacy; PMV = promotive voice behavior; PHV = prohibitive voice behavior; TS = team size.

**Table 5 ijerph-17-01162-t005:** Results of mediated and moderated analysis.

Variables	Main Effects Model								Full Model							
	Self-Efficacy		Psychological Safety		Promotive Voice Behavior		Prohibitive Voice Behavior		Self-Efficacy		Psychological Safety		Promotive Voice Behavior		Prohibitive Voice Behavior	
	R = 0.224		R = 0.275		R = 0.282		R = 0.334		R = 0.224		R = 0.275		R = 0.2790		R = 0.3305	
Paths	*b*	SE	*b*	SE	*b*	SE	*b*	SE	*b*	SE	*b*	SE	*b*	SE	*b*	SE
Direct effects																
Paradoxical leadership	−0.004	0.052 **	−0.287	0.198 **					−0.004	0.052 **	−0.287	0.198 **				
Self-efficacy					−0.143	0.243 *	−0.072	0.33 **					0.1474	0.2431 *	0.059	0.3299 **
Psychological safety					−0.066	0.065 **	−0.051	0.088 **					0.0716	0.0641 *	0.0449	0.0871 **
Moderating variables																
Team size					−0.029	0.76 **	0.081	0.103 *								
Interaction effects																
Self-efficacy X Team size													0.07	0.0186	0.003	0.0249
Psychological safety X Team size													0.0163	0.0248 *	0.0102	0.0301
Controls																
Employee age	0.008	0.031	0.116	0.117	0.1	0.07	−0.213	0.096	0.008	0.031	0.116	0.117	0.0985	0.0706	−0.2046	0.0957
Employee gender	−0.044	0.05 *	−0.216	0.189	−0.142	0.114 *	0.24	0.155	−0.044	0.05 *	−0.216	0.189	−0.1414	0.1147	0.2482	0.1556
Employee working years	0.013	0.033	−0.199	0.127	−0.063	0.077 *	0.159	0.105	0.013	0.033	−0.199	0.127	−0.0652	0.0773	0.1622	0.1048
Leader age	−0.041	0.031 *	0.013	0.117	0.002	0.072	−0.059	0.098	−0.041	0.031 *	0.013	0.117	−0.0031	0.0709	−0.04	0.0963
Leader gender	0.026	0.049	0.058	0.187	−0.031	0.112	0.09	0.151	0.026	0.049	0.058	0.187	−0.0281	0.1124	0.0969	0.1525
Leader working years	0.029	0.029	−0.136	0.108*	−0.113	0.066	0.006	0.089	0.029	0.029	−0.136	0.108*	−0.1104	0.066	−0.0058	0.0895

Note: * *p* < 0.05. ** *p* < 0.01.

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
