# Peer review of "How Does Paradoxical Leadership Affect Employees’ Voice Behaviors in Workplace? A Leader-Member Exchange Perspective"

_ijerph, 2020, doi:10.3390/ijerph17041162_

Round 1
Reviewer 1 Report
A potentially interesting and relevant study in the areas of leadership, exchange, and employee voice. This could be a welcome contribution to behavioural research in organisational studies.
However, there are some major revisions required before it can be assessed as an acceptable paper for publication.
Literature scoping and theoretical underpinnings
The authors apply a leader-member exchange (LMX) perspective. A much deeper review of this concept is required, and most importantly a justification for why they have not measured LMX in their survey. Is social exchange more sufficient and general to apply as a theoretical lens for their model? The authors have missed articles which have explored paradoxical leadership (PL) and its potential effects on employees through social exchange (see recent 2019 article on PL and employee resilience, for example). This is essential due to their emphasis on LMX. Not all PL studies are based on Yin Yang philosophy – be mindful of vast generalisations, show your understanding of PL by reflecting on the various perspectives Please provide a citation/s for claims made in Lines 51-53 Blackman & Sadler-Smith do not explicitly study the link between PL and voice as implied by the authors. Claims about existing studies need to be accurate in academic scholarship. Paradox theory and its core underpinnings needs more attention and elaborationWritten clarity and communication
The structure and framing of the whole paper needs work. This is particularly in relation to the way in which ideas are written and presented, in relation to the study aims. Opening paragraph relies too heavily on one source (Pearce et al., 2019) – broaden this to show the wider relevance of your paper to the existing literature. What are the implications of Pearce et al.’s study for PL and employee voice behaviours? Line 25 is ambiguous and needs to be made clearly relevant to the study – what exactly is the paradox in your example and why is it so important for your study to be in the opening paragraph? Line 31 comes as a surprise to the reader as to why the authors are suddenly discussing voice – needs to be integrated with opening paragraph. The ‘The’ opening the sentence needs to be removed. I strongly recommend this paper be carefully edited by a native English speaker. It does not read in a clear or engaging way. Line 97 – Lewis citation brackets need correcting Line 111 – Schwartz reference needs to be removed from in-text There is too much review and description of existing studies, and not enough integration of these studies with the authors’ own research aims and contributions. It reads like a literature review with no argument – the reader needs to be more convinced by the importance of the study and its positioning within existing research. Reference list needs more attention – correct formatting etc.Method and results
There needs to be a much stronger justification and explanation for choosing the 12 items from Zhang et al.’s PL scale. The original scale is over 20 items, how did you decide which items to retain/remove and why? Usually EFA and CFA is required before shortening an existing scale. The authors should provide more clarification around this – was the original scale used in the survey or were the 12 chosen beforehand (what were the decisions made here?)? Why are these 12 the most reliable items for a shortened measure? 12 items over 5 factors means some dimensions lack the widely established minimum of 3 items for factor. How can you show this is still a valid measurement? Please provide a justification for why LMX was not measured, particularly as a mediator. In the results section, please provide a visualised model with the results/coefficients shown for each pathway.The below references may help with various points raised above, i.e. paradox underpinnings, LMX and social exchange, PL and employee outcomes. These are starting points: a much more integrated foundation of literature needs to be applied in this paper. Currently literature is described but not synthesised for a coherent message.
Cunha, M. P. E., Fortes, A., Gomes, E., Rego, A., & Rodrigues, F. (2019). Ambidextrous leadership, paradox and contingency: evidence from Angola. The International Journal of Human Resource Management, 30(4), 702-727.
Kearney, E., Shemla, M., van Knippenberg, D., & Scholz, F. A. (2019). A paradox perspective on the interactive effects of visionary and empowering leadership. Organizational Behavior and Human Decision Processes.
Franken, E., Plimmer, G., & Malinen, S. (2019). Paradoxical leadership in public sector organisations: Its role in fostering employee resilience. Australian Journal of Public Administration.
Yi, L., Mao, H., & Wang, Z. (2019). How paradoxical leadership affects ambidextrous innovation: The role of knowledge sharing. Social Behavior and Personality: an international journal, 47(4), 1-15.
Waldman, D. A., Putnam, L. L., Miron-Spektor, E., & Siegel, D. (2019). The role of paradox theory in decision making and management research. Organizational Behavior and Human Decision Processes.
Walumbwa, F. O., Cropanzano, R., & Goldman, B. M. (2011). How leader–member exchange influences effective work behaviors: Social exchange and internal–external efficacy perspectives. Personnel Psychology, 64(3), 739-770.
Wayne, S. J., Shore, L. M., & Liden, R. C. (1997). Perceived organizational support and leader-member exchange: A social exchange perspective. Academy of Management Journal, 40(1), 82-111.
Reviewer 2 Report
Dear Authors,
Thank you very much for the opportunity to read your article. I find that your research relies on a powerful theoretical framework and on consistent study design. Besides, results are presented clearly and discussion paves the way for interesting conceptual and practical insights.
Although I recommend the publication of this paper, I also see at least two areas of improvement. Firstly, I think that your theoretical framework overlooks the interdependence between self-efficacy and psychological safety, which may "confound" the findings of your study. Secondly, beyond theoretical and managerial implications, authors should pay attention to broader "system-level" practical implications, in line with the aims and scope of IJERPH.
Once again, well done with this interesting piece of literature.
Yours,
The Reviewer
Reviewer 3 Report
Thank you for the opportunity to review this paper. The study is very interesting and has original aspects. I found that the paper is well-structured with the explanation of the main topics of the research with quite good consistent in the theoretical part. However I have some doubts about the methodology.
All of the hypothesis are negative statements: “There is no significant influence/effect...”. Does it make sense to study the variables that we assume that they are not correlated? I’m not sure. The study correlations between different variables which seem not to be correlated is very dangerous. If this (I mean study the variables, which we know or assume that they are not correlated) becomes a scientific trend, researchers can study everything in order to prove that the variables do not affect to each other. What about the contribution in the research development then? If the authors want to use negative statements, they should strongly justify the need of that research. For example, because of unreliability of the previous studies. In this paper the definitions and the concepts are well described, but the studies of the other authors in the topics are insufficient.
Additionally, the results are inconsistent with the conclusion. For example, the authors first indicated that “self-efficacy does not have significant indirect effect on relationship between paradoxical leadership and promotive voice behavior...” (line 472-473), and then “we find that paradoxical leadership has an indirect effect on employees’ voice behavior, through self-efficacy and psychological safety” (line 499-500). This is not clear.
To sum up, the theoretical part is quite well developed, but the changes in the empirical part should be considered for clarification of the results and the conclusion.
Round 2
Reviewer 1 Report
Thanks for your thorough responses to the earlier feedback. I am now confident this is a sound piece of research with a solid contribution.
Reviewer 3 Report
The authors are to be congratulated on the work that they have done to revise this paper. This is a much improved paper. However, I have still doubts about the methodology and then, the obtained results.
The authors wrote that “there is significant and positive correlation...” (line 493-497), but the mentioned correlations are less or slightly more than 0.1. This is not strong correlation, especially that the sample size is not large. I find that it difficult to conclude that two variables are correlated based on this results. There are positive (or negative) correlations between the variables, but they are insufficient to accept or reject the hypothesis. They can only support other results. This needs explanation and more conclusive results.
